# Impact of Frailty on Outcomes of First-Line Pembrolizumab Monotherapy in a Real-World Population with Advanced Non-Small Cell Lung Cancer

**DOI:** 10.3390/biology12020191

**Published:** 2023-01-26

**Authors:** Rocío Jiménez Galán, Elena Prado-Mel, Maria Alvarez de Sotomayor, Laila Abdel-Kader Martin

**Affiliations:** 1Clinical Unit of Pharmacy, University Hospital Virgen del Rocio, Avenue Manuel Siurot, 41013 Seville, Spain; 2Pharmacology Department, Faculty of Pharmacy, University of Seville, 41012 Seville, Spain

**Keywords:** pembrolizumab, non-small cell lung cancer, frailty

## Abstract

**Simple Summary:**

Real-world studies of immune checkpoint inhibitors (ICIs) in advanced non-small cell lung cancer (NSCLC) have shown worse outcomes in patients with poor Eastern Cooperative Oncology Group Stage Performance Status (ECOG PS). This index measures functional status but does not assess the cause. The ECOG PS scale is influenced by different aspects, such as the burden of the disease itself, the presence of comorbidities and the global frailty of patients. The influence of frailty on the efficacy of ICIs in patients with NSCLC has not been evaluated. In this study, we investigated the role of frailty on the clinical outcomes of first-line pembrolizumab in a retrospective cohort of 101 patients with advanced NSCLC. In our study, frailty determined based on indirect markers was identified as an independent predictor of overall survival (OS) and progression-free survival (PFS). Frailty assessment before starting antineoplastic therapy could be a useful tool for clinical decision making.

**Abstract:**

ICIs have been able to improve overall survival in advanced-stage lung cancer. The benefit of this therapy is limited in patients with poor ECOG PS. However, this scale is imprecise and can be influenced by different factors, such as frailty. Cancer patients have a high risk of frailty independently of age. In this observational, single-center, retrospective study, we investigated the effect of frailty on the effectiveness of pembrolizumab in first-line use in a cohort of 101 patients with metastatic NSCLC. Frailty was determined using a frailty score system developed by Sakakida et al. Univariate and multivariate analysis was performed to determine the prognostic role of frailty on OS and PFS. Median OS was significantly higher in patients with low frailty compared with intermediate and high frailty (23.8 vs. 7.0 and 1.8 months, respectively; *p* < 0.001). Median PFS was also significantly higher in patients with low frailty compared with intermediate and high frailty (10.5 vs. 3.9 and 1.6 months; *p* < 0.001, respectively). Frailty was the only variable that showed significant differences in OS and PFS. Multivariate analysis confirms frailty as an independent predictor of OS and PFS. Frailty assessment could help to select which patients are candidates for ICIs in NSCLC.

## 1. Introduction

In 2020, there were 2,206,771 (11.4%) new cases of lung cancer worldwide, and 1,796,144 (18%) deaths were caused by the disease [1]. The emergence of immune check-point inhibitors (ICIs) has substantially improved the prognosis of advanced-stage lung cancer, resulting in higher survival compared with conventional chemotherapy [2,3,4,5,6,7,8,9]. However, the benefit of this therapy is limited in some subgroups of patients, such as patients with poor Eastern Cooperative Oncology group performance status (ECOG PS) [10,11,12]. These findings have been shown in real-world data studies because clinical trials have excluded patients with PS ≥ 2 [13,14]. We recently published a study with real-life data in which patients with non-small lung cancer (NSCLC) and PS of 0-–1 treated with pembrolizumab in a first-line setting had better clinical outcomes than patients with ECOG PS ≥ 2 [15]. This scale is the one most commonly used in cancer patients to assess their disability level. However, this index measures functional status but does not assess the cause, and as a result, very different patients may be categorized with the same PS. The ECOG PS scale is influenced by different aspects, such as the burden of the disease itself, the presence of comorbidities and the global frailty of elderly people [16]. 

A person is considered to be fragile when there is a decrease in their biological reserve, generating greater vulnerability to stressful factors (cancer or treatments) and greater risk of adverse events (complications, dependence or death) [17,18]. According to recommendations agreed upon by a group of experts, all people with significant weight loss (>5%) due to chronic disease and all people older than 70 years should undergo frailty screening [19]. It is estimated that more than half of cancer patients are in a frail state. According to data published by the National Cancer Institute of the United States, the median age of lung cancer patients at diagnosis is 71 years, and more than 70% of cases are older patients [20]. Age-related changes in organ function and organ reserve can alter tolerability to and toxicity of pharmacotherapy. The higher prevalence of liver, kidney, heart and bone marrow diseases in the elderly also affects patient performance status (PS) by increasing polypharmacy and thus interactions that cause a higher risk of mortality in this population [21,22]. However, although frailty increases with age, it can occur at any age, particularly in patients with acute illness [23]. Frailty status is associated with an impaired immune system and a pro-inflammatory state [24]. The efficacy of ICIs has not been determined in frail patients with advanced NSCLC. This type of research is necessary to make clinical decisions and to select patients who will benefit from these therapies. In the present study, we analyze the influence of frailty on overall survival and progression-free survival in a cohort of patients with metastatic NSCLC treated with pembrolizumab in a first-line setting, using a frailty score system developed by Sakakida et al. [25].

## 2. Materials and Methods

### 2.1. Study Design and Data Collection

This was a cohort expansion of a previously published observational, single-center, retrospective study [15] that included all patients with advanced NSCLC with a PD-L1 Tumor Proportion Score (TPS) ≥ 50% and negative mutations of EGFR (epidermal growth factor receptor) or ALK (anaplastic lymphoma kinase) treated with pembrolizumab single-drug therapy as first-line treatment between 1 January 2017 and 31 December 2020. 

Demographic data (sex and age), history of smoking behavior, tumor histology, PD-L1 TPS, presence of central nervous system (CNS) metastasis, disease stage at diagnosis, ECOG PS, comorbidities and hematological parameters at the beginning of the administration of pembrolizumab were obtained from electronic medical records and oncology pharmacy registers.

Primary effectiveness endpoints were overall survival (OS), defined as the time from the start of pembrolizumab to death, and progression-free survival (PFS), defined as the time from the start of pembrolizumab treatment to progression or death. Those patients who did not present an event (progression and/or death) before the end date of the study were censored. Objective response rate (ORR) and stable disease (SD) were also analyzed. 

We determined the level of frailty for each patient at the start of pembrolizumab using the Frailty Scoring System (FSS) developed by Sakakida et al. (Appendix A). This tool considers indirect markers of frailty, including ECOG PS, Charlson comorbidity index (CCI) and neutrophil–lymphocyte ratio (NLR). The FSS categorizes frailty as low, intermediate or high. 

The NLR was calculated by dividing the neutrophil count by the lymphocyte count. We considered the blood count closest to the start of pembrolizumab.

### 2.2. Statistical Analysis

Statistical analysis was performed with the SPSS^®^ 24.0 statistical software. Continuous variables were expressed with mean or median measures and their corresponding measures of dispersion, standard deviation or interquartile range, respectively. The categorical variables were presented in frequencies and percentages. To compare continuous variables, the Mann–Whitney U test or Student’s *t*-test and Fisher’s exact test for categorical variables were used. The normal trend of the data was demonstrated with the Kolgomorov–Smirnov test. The Kaplan–Meier method was used for estimating the probability of survival. The long-rank test was used to determine the relationship between each variable and OS and PFS. Subsequently, the Cox regression model was performed with the variables that had shown statistical significance in the previous analysis. Hazard ratios and associated 95% confidence intervals were also calculated using the Cox proportional hazards model. Statistical significance was considered to exist when *p* < 0.05. The validity of the statistical model was verified with three statistics: likelihood ratio, Wald test, and log-rank. The assumption of proportionality of risks was also validated.

### 2.3. Ethics

Patients were anonymized with a code number in order to comply with the Organic Law on Data Protection 03/5 December 2018 (OPDL) and thus protect the confidentiality of patient data. Informed consent was not requested from the patients because it was a retrospective cohort.

## 3. Results

### 3.1. Baseline Population Characteristics 

In total, 101 patients were included in the study. Demographic and clinical characteristics are detailed in Table 1. Most patients were male and were smokers or ex-smokers. The median age was 67 years, and 65.3% were under 70 years of age. The most prevalent histology was adenocarcinoma (68.3%), followed by squamous cell (18.8%) and other histologies (12.9%). Almost all patients (97.7%) had stage 4 cancer at diagnosis and 15.8% presented brain metastases. PD-L1 TPS was ≥90% in 26.7% of patients.

### 3.2. Baseline Population Characteristics According to Frailty Scoring System

According to the FSS, at the beginning of pembrolizumab therapy, the level of frailty was low in 27.7% of patients, intermediate in 40.6% of patients and high in 31.7%. Most of the patients (90.1%) had a CCI score of 0–2, 37.9% of patients had an ECOG score ≥ 2 and 57.4% had an NLR ≥ 4.

Baseline characteristics according to the FSS are detailed in Table 2. No statistically significant differences were found in baseline demographic and clinical characteristics among patients with low, intermediate or high frailty. It should be noted that the proportion of elderly patients was similar in the three groups. Adenocarcinoma histology was less frequent in patients with high frailty, but this difference was not statistically significant.

### 3.3. Pembrolizumab Outcomes in Overall Population 

At the cut-off date (28 February 2022), with a median follow-up time of 30 months, 24 patients included in our study were alive (23.8%) and 86 (85.1%) had progressed, while 49.5% (n = 50) of patients died in the first three months of treatment with pembrolizumab. Median OS and PFS were 6.2 months (95% CI, 3.3–9.1) and 3.2 months (95% CI, 1.6–4.8), respectively. 

Regarding response rate, no patients in our cohort achieved a complete response, 28.7% presented a partial response, 17.8% showed stable disease and 26.7% showed progression disease. Tumor response could not be assessed in 26.7% of patients due to death or clinical progression before the first evaluation.

### 3.4. Impact of Frailty in Outcomes of Pembrolizumab 

Survival analysis of OS and PFS according to the FSS are shown in Figure 1. OS was significantly higher in patients with low frailty, with a median of 23.8 months compared with 7.0 months (95% CI, 3.8–10.2) and 1.8 months (95% CI, 1.2–2.6) in patients with intermediate and high frailty, respectively. Similarly, the median PFS in patients with low, intermediate and high frailty was 10.5 (95% CI, 0–21.7), 3.9 (95% CI, 2.1–5.7) and 1.6 months (95% CI, 0.9–2.3), respectively (*p* < 0.001). In the first 3 months, 84.3% (27/32) of patients with high frailty died compared with 46% (19/41) and 14% (4/28) of patients with intermediate and low frailty, respectively. 

Regarding ORR (Table 3), there were statistically significant differences according to the level of frailty. The PR rate was the same in patients with low and intermediate frailty. However, most patients with high frailty had PD as the best response (*p* < 0.001).

The results of univariate analysis for PFS and OS with the rest of the variables collected (sex, age, smoking history, histology, presence of brain metastases and PD-L1 TPS) did not show statistically significant differences (Table 4). 

The FSS was the only variable included in the Cox proportional-hazards model. The HRs of OS for intermediate and high frailty were 2.4 (95% CI, 1.3–4.5) and 6.3 (95% CI, 3.3–12.0), respectively (Figure 2a). The HRs of PFS for intermediate and high frailty were 2.0 (95% CI, 1.1–3.4) and 5.2 (95% CI, 2.9–9.4), respectively (Figure 2b).

## 4. Discussion

In this study, we explored the influence of frailty on the outcome of pembrolizumab in first-line therapy of NSCLC. Our data suggest that patients with higher frailty have worse clinical results. The median OS and PFS were significantly reduced as the FSS increased. To our knowledge, this is the first study that correlates frailty with the outcomes of immunotherapy in NSCLC. Some studies have shown reduced survival in frail patients with early stages of NSCLC [26,27,28,29]. A systematic review and meta-analysis showed poor prognosis in lung cancer patients with frailty or prefrailty [29]. However, this study includes different profiles of patients (early and late stages), and patients treated with ICIs were not evaluated. The study carried out by Sbrana et al. [28] evaluated the MPI (Multi-Pronostic Index) as a screening tool for 79 older adults with advanced or metastatic cancer eligible to receive immunotherapy. Authors found that patients with the highest MPI score experienced the worst overall survival, with a five-fold increased risk of mortality. In this work, the Short Physical Performance Battery (SPPB) was also used as a risk assessment of frailty in older adults. The prevalence of frailty or pre-frailty was high (63.3%), although, unlike in our study, the percentage of patients with ECOG PS of 2–3 was very low (11.4%). As in our study, frail patients treated with immunotherapy had worse outcomes. However, these results cannot be compared with ours because the population characteristics were very different. Sbrana et al. included older patients with several types of cancer and stages. 

There are various tools available to assess frailty in oncology patients, some of which have been validated in the elderly population with cancer, and more specifically in patients with lung cancer. Several systematic reviews attempting to analyze the impact of frailty in older patients with lung cancer have been published in recent years [30,31]. Among the most used tools in the different studies is the G-8 screening tool [32], developed to identify elderly cancer patients who may benefit of a comprehensive geriatric evaluation and thus maximize the health results that can be obtained through the approach chosen for your oncological pathology. The G-8 screening tool is an easy-to-use tool and is available online [3]. Other tools used are the Vulnerable Elderly Survey 13 (VES 13), the Comprehensive Geriatric Assessment (CGA) or the Fried Frailty Index (FFI), among others. Our study, being retrospective, limited the use of self-report or prospective tools, which is why the use of the abovementioned tools to assess the degree of frailty of the patients included in the study was rejected. The Sakakida tool is a combined variable that uses three indirect indicators of frailty [25]. 

Other authors have evaluated the influence of these factors separately. ECOG PS has been identified as a prognostic factor of survival in patients with NSCLC treated with pembrolizumab in a first-line setting [33,34]. In a previously published study [15], we found that patients treated with pembrolizumab with ECOG PS ≥ 2 had a significantly shorter OS and PFS than those with ECOG PS of 0–1, with a median OS of 2 months and 18.9 months, respectively. However, the use of the ECOG as a prognostic marker is insufficient, and it is necessary to investigate other aspects related to the patient. The FSS is a more complete tool in which, in addition to the ECOG PS, other important factors, such as the NLR and comorbidities, are considered. 

Other real-world data studies have obtained similar outcomes to ours. In the same way, the prognostic effect of the baseline NLR on OS and PFS has been proven in patients treated with ICIs [35,36], and recently, Alessi et al. [37] published a paper stating that patients treated with first-line pembrolizumab with PD-L1 TPS ≥ 50% and a high NLR had a significantly lower ORR (52.4% vs. 24.7%, *p* < 0.001) and median PFS (10.4 vs. 3.4 months, HR: 0.48 (IC 95%: 0.35–0.66, *p* < 0.001)) than patients with a low NLR. Other authors have found an association between high basal NLR levels with resistance to treatment with ICIs [38,39]. These results may be in line with our results, since the disease progression rate in the high-frailty cohort was 85%, and this could be explained by the high NLRs in this cohort. In this sense, the NLR could be considered as a confounding factor for frailty in our population. However, frailty is associated with a pro-inflammatory state characterized by an elevation of inflammatory markers, including peripheral white blood cells.

The influence of comorbidities on the efficacy of ICIs has also been evaluated by some authors. However, no conclusive results have been found in this regard [40,41,42]. In this research, we determined the comorbidities of our cohort through the CCI. Overall, our population had low CCI scores, and their effects on frailty could be limited in our study. In any case, it is difficult to compare with other studies because there is wide variability in the classification of patients according to the CCI score. We categorized patients into two groups according to scores 0–2 and ≥3, while other authors classified patients with CCI scores ≥ 1 and <1 or 0, 1, 2 and ≥3.

The proportion of older patients (>70 years of age) was low, which could explain the low CCI score in our population. There are different definitions of frailty, although most of them agree that frailty increases with aging. However, in patients with chronic illnesses, frailty can occur at any age [43]. In our study, frailty was not associated with age. On the other hand, it should be noted that most patients with high frailty died in the first three months after the start of therapy. In the pivotal clinical trial Keynote-024 [8], the median time to respond to pembrolizumab was 2.2 months, with the survival curve of the pembrolizumab arm starting to separate from the chemotherapy arm after 2 months of treatment. Due to the poor short-term prognosis in frail patients, the benefit of immunotherapy in these types of patients may be limited.

Based on these results, the frailty of patients could play a key role in therapeutic decision making. Due to the mechanism of action of these drugs and the marked relationship of frailty with the immune system, this tool could be applied to select patients who are candidates for treatment with immunotherapy. However, to confirm these results, it is necessary to carry out prospective studies with validated frailty tools. In addition, more studies are needed to compare the FSS with other frailty assessment tools and find out the best tool to apply in this population.

This study has some limitations, such as retrospective design and, as a consequence, the use of indirect markers of frailty. Nevertheless, given that the outcome variables assessed are relevant endpoints such as OS and PFS, and the components assessed in frailty are routinely collected variables in clinical practice, the bias of those variables observed is unlikely. Furthermore, the Sakakida tool is not a widely validated tool for measuring frailty.

## 5. Conclusions

In conclusion, high frailty, evaluated through indirect markers of frailty, significantly reduces survival in patients with advanced NSCLC who received first-line therapy with pembrolizumab. Frailty status assessment in cancer patients before treatment could support decision making with individualized cancer treatment planning to avoid under- or overtreatment. 

## Figures and Tables

**Figure 1 biology-12-00191-f001:**
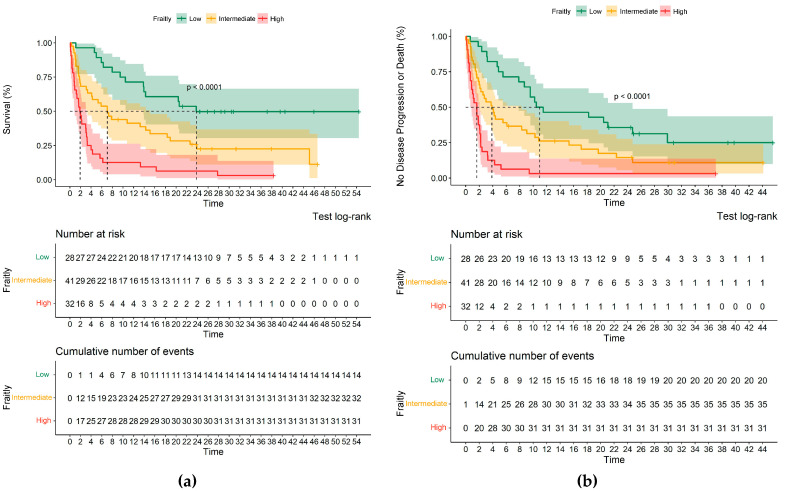
Kaplan–Meier analysis of overall survival and progression-free survival. Figure (**a**,**b**) show Kaplan–Meier estimates of overall survival and progression-free survival stratified according to the level of frailty based on the Frailty Scoring System.

**Figure 2 biology-12-00191-f002:**
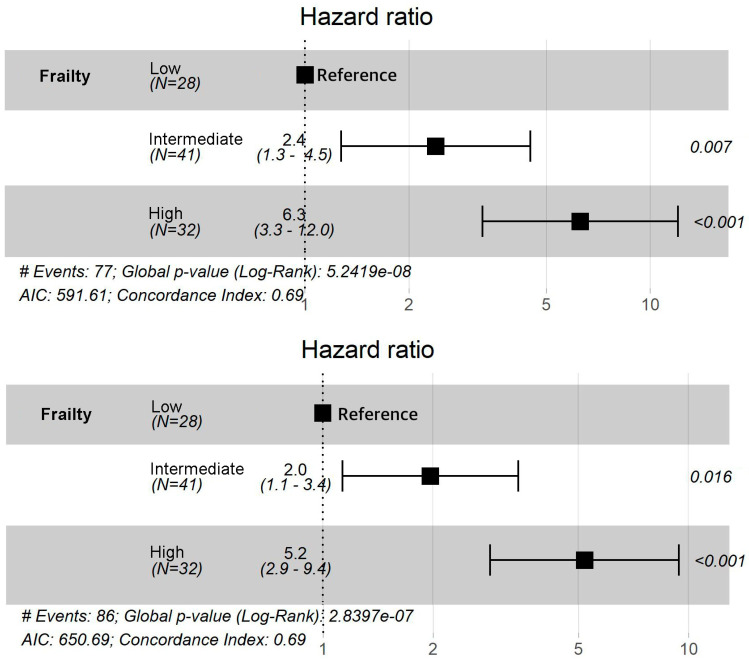
Cox proportional-hazards model for Overall survival and progression-free survival. Figure (**a**,**b**) show hazard ratios and 95% confidence intervals for overall survival and progression-free survival in subgroups defined according to level of frailty based on the Frailty Scoring System.

**Table 1 biology-12-00191-t001:** Baseline demographic and clinical characteristics of patients.

Characteristics	Values, n (%)
Sex Male Female	
	75 (74.3)
	26 (25.7)
Age <70 years of age ≥70 years of age	
	63 (62.4) 38 (37.6)
Smoking history Yes No	
	93 (92.1) 8 (7.9)
Histology Adenocarcinoma Squamous NSCLC poorly differentiated Others	69 (68.3) 19 (18.8) 9 (8.9) 4 (4.0)
Disease stage IIIB IV	2 (2.3) 86 (97.7)
Brain metastases Yes No	16 (15.8) 85 (84.2)
PD-L1 TPS% <90% ≥90%	74 (73.3) 27 (26.7)
ECOG PS 0 1 2 3	20 (19.8) 43 (42.6) 30 (29.7) 8 (7.9)
CCI 0–2 ≥3	91 (90.1) 10 (9.9)
NLR ≥4 <4	58 (57.4) 43 (42.6)
Frailty Scoring System Low Intermediate High	28 (27.7) 41 (40.6) 32 (31.7)

NSCLC: non-small cell lung cancer; ECOG PS: Eastern Cooperative Oncology Group performance status; PD-L1: programmed cell death ligand 1; TPS: tumor proportion score; CCI: Charlson Comorbidity Index; NLR: Neutrophil–Lymphocyte Ratio.

**Table 2 biology-12-00191-t002:** Baseline classification of patients according to the Frailty Scoring System.

Characteristics	Frailty Scoring System			*p*-Value
	Low n (%)	Intermediate n (%)	High n (%)	
Sex Male Female	21 (75) 7 (25)	29 (70.7) 12 (29.3)	25 (78.1) 7 (21.9)	0.769
Smoking history	25 (89.3)	38 (92.7)	30 (93.8)	0.801
Histology Adenocarcinoma Non-adenocarcinoma	20 (71.4) 8 (28.6)	31 (75.6) 10 (24.4)	18 (56.3) 14 (43.8)	0.193
PD-L1-expression levels ≥90% <90%	7 (25) 21 (75)	11 (26.8) 30 (73.2)	9 (28.1) 23 (71.9)	0.963
Elderly No (age < 70years) Yes (age ≥ 70 years)	16 (57.1) 12 (42.8)	26 (63.4) 15 (36.6)	21 (65.6) 11 (34.4)	0.783
Brain metastases Yes No	5 (17.9) 23 (82.1)	4 (9.8) 37 (90.2)	7 (21.9) 25 (78.1)	0.350

PD-L1: programmed cell death ligand 1.

**Table 3 biology-12-00191-t003:** Rate of response according to Frailty Scoring System.

Response	Low FSI	Intermediate FSI	High FSI
PR, n (%)	11 (40.7)	16 (50.0)	2 (13.3)
SD, n (%)	12 (44.4)	6 (18.8)	-
PD, n (%)	4 (14.8)	10 (31.3)	13 (86.7)

PR: partial response; SD: stable disease; PD: progression disease; FSS: Frailty Scoring System.

**Table 4 biology-12-00191-t004:** Univariate analysis for progression-free survival and overall survival.

	Progression-Free Survival		Overall Survival	
Variable	Median (95% CI)	*p*-Value	Median (95% CI)	*p*-Value
Sex Female male	3.2 (0.7–5.7) 3.2 (1.4–4.9)	0.630	4.2 (0–13.8) 6.3 (3.8–8.7)	0.805
Age <70 ≥70	3.2 (1.5–4.9) 3.9 (1.1–6.7)	0.829	5.7 (1.9–9.4) 7.0 (3.9–10.1)	0.934
Smoking history No Yes	4.4 (0–14.7) 3.2 (1.6–4.8)	0.566	10.8 (0–31.3) 5.9 (3.5–8.3)	0.237
Histology Adenocarcinoma Non-adenocarcinoma	4.2 (2.3–6.2) 2.1 (1.9– 2.3)	0.303	7.0 (2.2–11.9) 3.3 (0–7.0)	0.282
PD-L1 TPS <90% ≥90%	3.1 (1.2–5.0) 3.3 (0.7–5.8)	0.554	6.2 (2.8–9.6) 5.3 (0–11.7)	0.742
Brain Metastasis No Yes	3.7 (1.9–5.6) 2.2 (0.04–4.3)	0.164	7.0 (1.9–12.2) 4.1 (2.2–5.9)	0.205
Frailty Scoring System Low Intermediate High	10.5 (0–21.2) 3.9 (2.1–5.7) 1.6 (0.9–2.3)	<0.001	23.8 (–) 7.0 (3.8–10.2) 1.8 (1.2–2.6)	<0.001

PD-L1: programmed cell death ligand 1.

## Data Availability

The data presented in this work have not been previously published.

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
