# Peer review of "Impact of Frailty on Outcomes of First-Line Pembrolizumab Monotherapy in a Real-World Population with Advanced Non-Small Cell Lung Cancer"

_biology, 2023, doi:10.3390/biology12020191_

Round 1

Reviewer 1 Report

The reviewed article presents the results obtained from the analysis of data obtained retrospectively from a cohort of patients suffering NSCLC and treated with pembrolizumab. The authors find a correlation between frailty of patients determined by FSS and the clinical outcome measured as OS and PFS. The results are clearly described and presented; however, a few comments should be addressed:

-          - The authors could describe in more detail what did the cohort expansion consist of and how it was done. Was a sample size calculator used to determine the number of patients to be included? Do authors consider the sample size sufficient, especially considering the % of patients with certain characteristics such as ECOG 2, female sex, or age 70?

-          - Can authors shed light on the processes or factors associated to frailty that could be responsible for the worse response of these patients to ICI’s? Which mechanism can be explored or proposed in order to clarify how frailty affects the mechanism of action or the response to these drugs?

-         -  In the discussion section, the authors could emphasize on the applicability of their results. Specifically, which benefits do they consider the findings that they show may have for patient care?

-          - Which are the advantages of using FSS as a tool to classify patients based on their frailty? Which frailty assessment tool do authors consider more appropriate? How is FSS a better choice compared to other scales which also consider frailty, such as ECOG PS?

-         -  Can these findings be extended to other types of chemotherapies other than pembrolizumab?

Reviewer 2 Report

More detailed discussion of the choice of frailty score is needed.  Is this a validated score in oncology patients apart from the Sakakida study? Are there frailty scores that have been used more widely in oncology patients? The conclusion refers to 'frailty' in some sentences when I think the authors mean to refer to high frailty e.g. line 247. It is important to discuss, as the authors do, the high mortality rate in patients with poor performance status across other real-world studies.  An 85% PD rate in the high frailty cohort does however question whether the measure of frailty is perhaps also a measure of aggressive disease with poor biology e.g. the use of the NLR may be describing the disease more than frailty.
